# New Onset Cardiac Murmur and Exertional Dyspnea in an Apparently Healthy Child: A Rare Localization of Obstructive Myxoma in the Right Ventricle Outflow Tract without Pulmonary Embolization—A Case Report and Literature Review

**DOI:** 10.3390/ijerph191912888

**Published:** 2022-10-08

**Authors:** Carolina D’Anna, Alberto Villani, Antonio Ammirati, Paola Francalanci, Laura Ragni, Giulia Cecconi, Aurelio Secinaro, Marcello Chinali, Antonella Santilli, Paolo Guccione, Lorenzo Galletti, Gianluca Brancaccio

**Affiliations:** 1Department of Cardiac Surgery, Cardiology and Heart and Lung Transplant, Bambino Gesù Children’s Hospital, IRCCS, 00165 Rome, Italy; 2Department of Emergency, Admission and General Pediatrics, Bambino Gesù Children’s Hospital, IRCCS, 00165 Rome, Italy; 3Department of Pathological Anatomy, Bambino Gesù Children’s Hospital, IRCCS, 00165 Rome, Italy; 4Advanced Cardiovascular Imaging Unit, Department of Imaging, IRCCS Bambino Gesù Children’s Hospital, 00165 Rome, Italy

**Keywords:** tumor, heart, children, right ventricle, echocardiogram, embolism, Carney complex

## Abstract

Myxomas are slowly growing benign neoplasms which are rare in children. Up to 80% can be located in the left atrium and generate symptoms such as embolism, cardiac failure, fever and weight loss. Rarely, myxomas can be detected in the right ventricle outflow tract, causing arrhythmias, pulmonary emboli and sudden death. We report the case of a 13-year-old healthy child brought to the Emergency Department (ED) of the Children’s Hospital Bambino Gesù, Rome, for recent dyspnea, chest pain on exertion and new onset cardiac murmur. Patient underwent medical examination and echocardiogram with the finding of a rounded and lobulated voluminous mass in the right ventricle outflow tract (RVOT) which caused severe obstruction. The contrast computed tomography (CT) scan confirmed the presence of a heterogeneously enhancing soft-tissue mass occupying the RVOT with no evidence of pulmonary embolization. The mass was surgically excised, and the pathologic examination confirmed our suspicion of myxoma. Our experience suggests that myxoma can have mild clinical symptoms, the presentation may be non-specific, and diagnosis can be a challenge Careful examination and a diagnostic imaging workup, primarily with the transthoracic echocardiogram, are needful to make a rapid differential diagnosis and to better manage surgical treatment and follow-up.

## 1. Introduction

Primary cardiac tumors are extremely rare (incidence less than 0.1%), whereas metastatic involvement of myocardium and pericardium can be more frequent [1,2]. Clinical manifestations of cardiac tumors can be completely absent, and often the diagnosis is incidental. The severity of signs and symptoms depends on the localization and not on the histopathology, and diagnosis can be challenging because the symptoms tend to mimic other cardiac and pulmonary diseases.

In adults, benign cardiac tumors represent more than 75% of cardiac tumors, and myxomas are the most frequent type, especially in middle-aged women, whereas in children they are rare [3,4].

Rhabdomyomas followed by teratoma, fibroma and hemangioma are predominant in childhood and fetal life [5,6].

## 2. Case Report

We report a case of a 13-year-old healthy male adolescent brought to the Emergency Department (ED) of the Children’s Hospital Bambino Gesù, Rome, for mild dyspnea, chest pain on exertion and new onset cardiac murmur. No syncope or palpitations both at rest and/or during exertion were reported.

Family history was negative for congenital heart disease, cardiomyopathy, sudden death and thrombophilia. Symptoms began three weeks before ED presentation, worsening with intense physical activity. Patient reported no fever or headaches, no history of recent infections, weight loss, venous catheters and/or tattoos. Upon arrival at the ED, the patient was alert and responsive, and completely asymptomatic at rest.

His vital signs upon presentation showed a normal body temperature, heart rate of 95 beats per minute, blood pressure of 96/72 mmHg and 100% oxygen saturation. The blood exams showed mild anemia (hemoglobin 12.9 g/dL) and augmentation of GPT-transaminases (43 U/L). Coagulation parameters, renal function tests, C-Reactive Protein and high-sensitivity troponin were normal, but NT-proBNP was elevated (4019 pg/mL). Chest roentgenogram was unremarkable.

Physical exam was characterized by regular heart rate and rhythm, interrupted by few extra systolic beats, with normal S1 and S2 and cardiac systolic murmur graded 3/6 Levine (highest at pulmonary focus) without gallops and rubs. The peripheral pulses were present and symmetrical. There was mild hepatomegaly.

ECG revealed sinus rhythm with normal PR and QTc intervals, right bundle block with ventricular repolarization abnormalities in the right leads and isolated monomorphic premature ventricular contractions (PVCs), often organized in bigeminy rhythm (Figure 1).

Echocardiogram identified a rounded and lobulated voluminous (3.5 × 4 cm) mass with clear margins and inhomogeneous content, adherent to the infundibular septum through a pedicle. The right ventricle outflow tract (RVOT) showed a severe obstruction with a maximum instantaneous systolic gradient of 80 mmHg, sparing the pulmonary valve (Figure 2 and Figure 3). The right atrium (Area 23.5 cm^2^) and ventricle (tele-diastolic area was 23 cm^2^ and tele-systolic 19 cm^2^) were dilated with a severe systolic dysfunction (Fractional Area Change of 18.6%).

Contrast CT scan confirmed the presence of a heterogeneously enhancing soft-tissue mass occupying the RVOT with no evidence of pulmonary embolization (Figure 4).

Mass was completely surgically excised.

Patient underwent a longitudinal median sternotomy and thyme retraction with opening and suspension of pericardium on cardiopulmonary bypass in normothermia (by arterial cannulation in ascending aorta and bicaval venous). By means of aortic clamping and induction of cardioplegia, it was possible to approach the mass through the right atrium. The mass (4 cm × 4 cm) had a pedicle originating in the right outflow a few mm from the pulmonary valve and jutting towards the pulmonary valve itself, obstructing the right ventricle outflow. The rather gelatinous mass was removed with the pedicle, including part of the muscle in order to completely remove the lesion.

The right atrium was closed with Prolene and there was a resumption of cardiac sinus rhythm with a good weaning from the extra corporeal circulation. Two mediastinal drains and two atrial electrodes were placed.

Sternosynthesis was performed in steel and skin and subcutis were sutured with absorbable points.

Pathologic examination confirmed the diagnosis of myxoma (Figure 5).

No complications during or after surgery were reported, and the intraoperative transesophageal echocardiography (TEE) showed no residual lesion and RVOT obstruction was totally relieved.

Patient was transferred to the Cardiac Intensive Care Unit and monitored for 24 h. He was finally discharged six days after admission. Right ventricle dimensions and function improved before discharge, with normal values.

During the follow-up, the patient showed good clinical conditions and no therapy was administered.

Serial echocardiograms did not show complications or recurrence of disease at 1, 3, 6, 12 and 24 months after surgery.

At two months, one and two years of follow-up, we performed a 24-hour Holter ECG monitoring which showed no signs of arrhythmias. The boy also underwent a treadmill exercise test without any symptoms, ECG anomalies and/or arrhythmias. A genetic examination was also performed, and it did not show a Carney complex syndrome association.

The ECG shows PVCs arising likely from the pulmonary artery/free wall of the right ventricular outflow tract. An LBBB configuration with inferior axis is seen with late precordial transition after V3 and QS in V1 e rS in lead I.

Transthoracic echocardiography shows the presence of a mobile intracardiac mass protruding into the right outflow, attached to the infundibular septum by a peduncle.

Transesophageal echocardiogram through the mid-esophageal 30° and 60° view, before and after surgery resection of myxoma, shows the complete resolution of RVOT obstruction.

Post-contrast CT confirms the presence of an ovoidal mass with smooth margins anchored to the infundibular septum posteriorly, determining significant RVOT obliteration as shown in this frontal and later multiplanar reformatting view. The lesion has small tissue density with mild enhancement and patchy distribution. There was no CT evidence of mass infiltration, pericardial effusion or pulmonary embolism.

The surgical mass was a whitish ovoid nodule, 3 cm in diameter, with a smooth surface (Figure 5A). Histological examination shows in a myxoid background cell proliferation, mostly loose, multifocally more densified, of single cells or in cords with a low mitotic index, in the absence of atypia (Figure 5B). Myxoma cells are characteristically located around the blood vessels (Figure 5B, insert). Bleeding areas are interposed. The tumor is covered by a fibrous, smooth, pseudo capsule. At the level of the implant base, myocardial cell is interposed in neoplastic proliferation.

## 3. Narrative Review of Literature and Discussion

Myxomas are slowly growing benign neoplasms which are particularly rare at pediatric age with a prevalence of 0.0017% and 0.19% in the autopsy study [7].

Histological analysis describes a mucopolysaccharide stroma with scattered cells capable of producing a vascular endothelial growth factor which contributes to growth and angiogenesis [8,9]. They have a gelatinous consistency and present at different sizes. In most cases, they can have a smooth surface and greater dimension, whereas in one-third, they have a friable or villous surface with a high embolization risk. These tumors can present two kinds of morphology: polypoid with obstructive features in approximately two-thirds of cases and papillary, with more embolization risk in one-third of cases because of their loose consistency and fragility.

Myxomas are located predominantly in the left atrium [10]. Patients mainly present signs and symptoms of heart failure for impairment of mitral valve function and secondary pulmonary hypertension, but there may also be serious neurological involvement (30–40% of cases) for embolism of the tumor or thrombi fragmentation in systemic circulation. Fever and weight loss are due to the release of cytokines, especially interleukin-6 (IL-6) and growth factors [11,12].

In a few patients, it is possible to listen for a typical “tumor plop” during the clinical exam. Very infrequently, myxomas are localized in right ventricle outflow (3–4%) with signs and symptoms of right heart failure, pulmonary embolism, arrhythmias, and sudden death [13,14,15,16].

They can be associated with Carney complex syndrome in 7% of cases with a predominant left-side localization. This inherited, autosomal dominant disorder is characterized by the presence of skin pigmented alterations (lentigines and blue nevi on the face, neck and trunk) and recurrent atrial and extracardiac myxomas, endocrine tumors and PRKARIA gene mutation [17].

Echocardiography represents the first noninvasive, easily accessible and low-cost imaging approach to confirm myxoma and to define localization, possible valves invasion and worsening mobility. Furthermore, transesophageal echocardiography (TEE) can provide superior diagnostic information for the pre-operative and postoperative evaluation.

Cardiac magnetic resonance (CMR) and computed tomography (CT) are further noninvasive and high-resolution imaging methods. Usually CMR is preferred, if immediately available or not contraindicated, for the ability to differentiate the type of tumor through the T1- and T2-weighted sequences.

To avoid risk of embolization, myxomas do not require biopsy if diagnosis is clear with a noninvasive imaging method.

Surgical resection represents the definitive treatment with a rapid postoperative recovery, with mortality rate under 5% and recurrence rate of 2–5% in sporadic cases and 12–22% in genetic forms [18,19,20].

Postoperative rhythm disturbance can appear in a minority of cases (atrial arrhythmias or atrio-ventricular conduction abnormalities) [13].

The literature review shows scarce data regarding right ventricle myxomas (only 23 cases in Japan have been described), and previous studies consist mainly of collections of case reports with left-side heart localizations [21,22].

Similar to ours, five cases of right ventricle myxomas have been previously reported in adults (27-, 17- and 30-years-old women and 49-, 26- and 46-years-old men). All cases were characterized at onset with pulmonary embolism and pulmonary hypertension. These cases demonstrated how important it is to investigate the source of pulmonary emboli, particularly in young patients with no known risk factors [23,24,25,26,27].

Right-side myxoma signs and symptoms can mimic deep vein thromboembolism after predisposing situations such as recent orthopedic surgery and immobilization.

In fact, in a young woman aged 26 years old, without risk factors, who experienced acute pulmonary embolism, a multicentric right atrium and ventricle myxoma were identified as cause for fragments embolization after a long diagnostic process [28].

Pulmonary embolism without deep vein thrombosis evidence at the venous Doppler or ultrasound imaging should become suspicious of possible right atrial or bi-atrial myxomas. Many case reports in adults have associated this location with pulmonary embolism complications [29,30,31].

Sometimes surgical intervention may leave the issue unresolved and the tumor can recur. Recurrence rate is rare (<3%), but increases between 12% and 22% in familiar forms and when associated with risk factors such as multicentricity, young age, familiar cases and incomplete surgical resection.

A recent case reported a 46-years-old patient with a right ventricle myxoma in the context of Carney complex syndrome. The patient had skin lesions, pituitary adenoma and Sertoli cell tumor and, after two years from surgical resection, he experienced a new mass in the right atrium [32].

A recurrence of right ventricle myxoma with pulmonary embolization was reported also in a 28-years-old patient because of an incomplete resection due to the adherence of the tumor to the tricuspid valve [33].

Moyassakis and Segal have described similar cases of recurrence after incomplete surgical resection in 21- and 34-year-old men [34,35].

Tatebayashi et al. reported a case of a 76-years-old male with a right ventricle myxoma involving the pulmonary artery extending to bilateral main pulmonary arteries with a severe occlusion [22].

In our case, the clinical presentation was unusual considering the nuanced symptomatology and the appearance of good health, which can make the diagnosis insidious. Moreover, unlike the cases reported in the literature, in our experience there were no signs and symptoms of pulmonary embolism, and this case represents one of the only two cases reported in early childhood with right ventricle myxoma [36].

In 1980, Gonzales described a massive pulmonary embolism in a child with right ventricle myxoma with total occlusion of the left main pulmonary trunk and a branch of the right pulmonary artery. Unfortunately, the surgical procedure was unsuccessful and the patient died [37].

New onset systolic murmur, dyspnea and chest pain on exertion were of paramount importance for us to drive the diagnosis and prompt management, but it was essential to immediately perform a transthoracic echocardiography (TTE) to detect the tumor and its localization.

The child underwent echocardiogram for referred symptoms, but also for the presence of new systolic murmur on pulmonary focus during cardiac auscultation. The ECG revealed ventricular repolarization anomalies suggesting right pressure overload. Moreover, the child was referred to our attention by colleagues from a peripheral center who had already performed an echocardiogram with diagnosis of intracardiac thrombus adhering to pulmonary valve. Our decision to perform echocardiogram was motivated by both clinical suspicion and the need to confirm or rebut the previous diagnosis.

Transesophageal echocardiography (TEE) drove surgical intervention to individualize the junction point.

Both TTE and TEE played important roles along with CT, being supplemental in the diagnostic process and differential diagnosis.

In fact, myxoma of right ventricle could be mistaken for endocarditic vegetation, embryonic remnant, thrombi, lipoma and non-myxomatous neoplasm. We promptly excluded these differential diagnoses. Particularly, the patient did not have signs or symptoms of sepsis, and had no thrombotic risk factor or history and evidence of extra cardiac tumor for an eventual metastatic cardiac involvement. Moreover, echocardiography can help in differential diagnosis. We excluded thrombus, commonly situated in an apical position or on the lateral wall, due to the absence of spontaneous echo contrast and perfusion. We did not consider malignant tumors due to the absence of mass high vascularization and greater contrast enhancement than adjacent myocardium. Rhabdomyoma was not contemplated due to the lack of bright spots and the observation that the tumor did not have the classic encapsulated cystic mass appearance as teratoma or encapsulated subepicardial mass as lipoma. Myxoma appeared to us the probable diagnosis.

## 4. Conclusions

This case represents an unusual presentation of a rare disease. Our experience suggests that myxoma can have mild clinical symptoms and the diagnosis can still be misleading Careful examination and early diagnostic imaging workup, especially with accurate echocardiogram, are needed in order to make a differential diagnosis, exclude massive pulmonary embolism, predict recurrence risk factors and improve surgical treatment and follow-up.

Myxoma associated with pulmonary embolism can range from asymptomatic to a life-threatening condition with high morbidity rate or immediate death when there is a massive diffusion, as reported by Tatebayashi and Gonzales [22,37].

In our case, the prognosis was good because the lack of pulmonary embolism due to fibrous, smooth, pseudo capsule tumor cover did not further compromise the patient’s clinical condition and did not prolong the duration of surgery, reducing surgical risks (especially bleeding) and recovery times.

## Figures and Tables

**Figure 1 ijerph-19-12888-f001:**
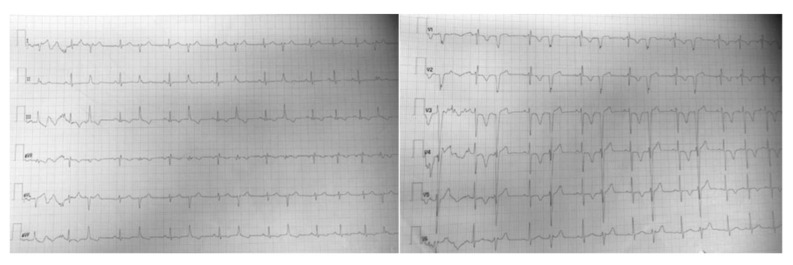
ECG.

**Figure 2 ijerph-19-12888-f002:**
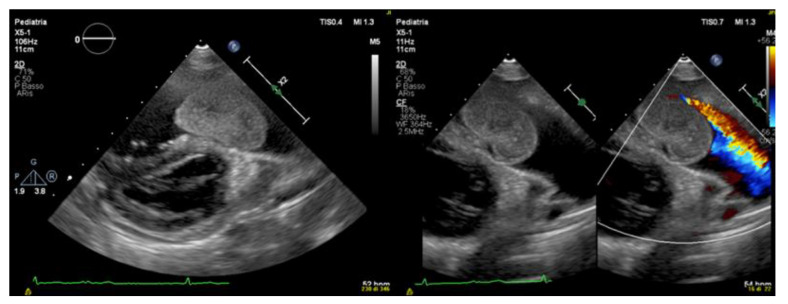
Color Doppler transthoracic echocardiography.

**Figure 3 ijerph-19-12888-f003:**
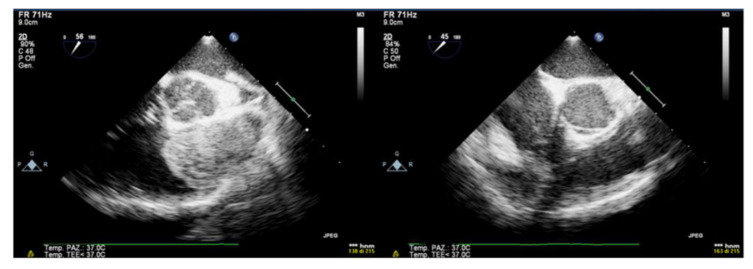
Color Doppler transesophageal echocardiography.

**Figure 4 ijerph-19-12888-f004:**
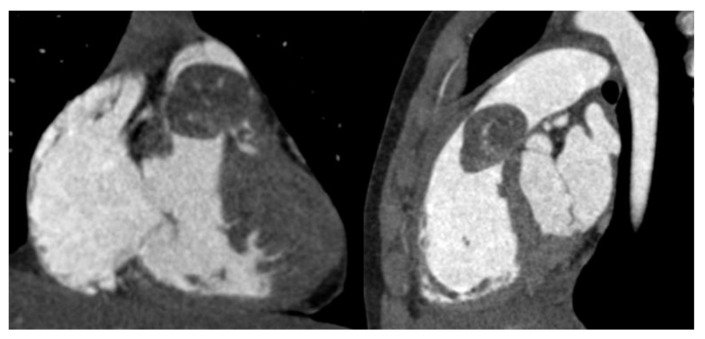
Computed tomography.

**Figure 5 ijerph-19-12888-f005:**
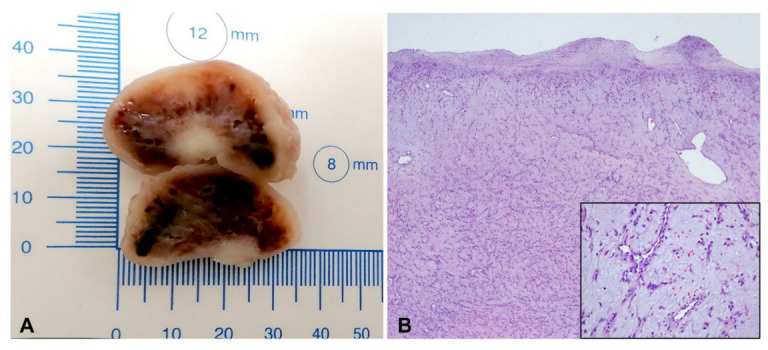
Resected infundibular myxoma (**A**) Section of myxoma biopsy (**B**) Hystological analysis.

## Data Availability

Individual participant data that underlie the results reported in this article will be available upon request after de-identification.

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
