# Peer review of "New Onset Cardiac Murmur and Exertional Dyspnea in an Apparently Healthy Child: A Rare Localization of Obstructive Myxoma in the Right Ventricle Outflow Tract without Pulmonary Embolization—A Case Report and Literature Review"

_ijerph, 2022, doi:10.3390/ijerph191912888_

Round 1

Reviewer 1 Report

Comments to authors,

Thank you very much for submitting an unusual case to this journal. I would like to hear the opinions of the authors on a few points.

#1. I wish there was a more detailed explanation about the operation.

(e.g., location of the myxoma)

#2. Only the presence or absence of embolism was mentioned in this case, and it seems necessary to explain what the prognosis is compared to the cardiac myxoma case with embolism.

#3. Not all patients undergo echocardiography. In this case, I think that a clear reason for performing echocardiography should be presented.

#3. How about combining the overlapping contents of the introduction and the narrative review into one place?

#4. I think peripheral embolism is an awkward term. Isn't that pulmonary embolism? (#226)

#5. It seems that the conclusion should be written concretely. All Children who complain of dyspnea can't be examined with the  echocardiography. 

In this case, the reason for considering echocardiography is not specifically stated.

Best regards,

Reviewer 2 Report

The authors present a case of a rare disease with a not-so-typical presentation. The case is interesting and well illustrated. Introduction and discussion give an accurate summary of the literature published on the topic. 
I have a few minor comments that may hopefully benefit the paper:

- Was the patient asked about recent weight loss at presentation?

- Premature ventricular contractions (PVCs) may be a preferable terminology compared to ventricular extra systoles. Also, what was the morphology of the PVCs at ECG (line 105)? Was it compatible with a PVC originating from the RVOT

- A brief description of the surgical excision technique may be added: what was approach? Was the procedure carried out off-pump or on-pump?

- Did right ventricular volumes and function normalize at the control echocardiograms?

- Was the myxoma smooth or villuos? A macroscopic description of the specimen can be added in the description of Figure 4. If the myxoma was smooth, this may be a possible explanation for the absence of pulmonary emboli and can be added in the discussion (line 194-196)

- English can be corrected or improved in many instances: incidental vs. incidentally (line 39), challenging vs. challenge (line 41), more than vs. over the (line 43), higher vs more (line 53), "the" (line 177), context vs contest (line 181), challenging vs. challenge (line 194) and so on
